# “Our Mothers Have Handed That to Us. Her Mother Has Handed That to Her”: Urban Aboriginal and Torres Strait Islander Yarning about Community Wellbeing, Healthy Pregnancies, and the Prevention of Fetal Alcohol Spectrum Disorder

**DOI:** 10.3390/ijerph20095614

**Published:** 2023-04-23

**Authors:** Vivian Lyall, Sonya Egert, Natasha Reid, Karen Moritz, Deborah Askew

**Affiliations:** 1Medical School, General Practice Clinical Unit, Level 8 Health Sciences Building, Royal Brisbane & Women’s Hospital, The University of Queensland, Herston, QLD 4029, Australia; 2Southern Queensland Centre of Excellence in Aboriginal and Torres Strait Islander Primary Health Care, Queensland Health, P.O. Box 52, Inala, QLD 4077, Australia; 3Child Health Research Centre, University of Queensland, Brisbane, QLD 4101, Australia

**Keywords:** urban Aboriginal and Torres Strait Islander communities, fetal alcohol spectrum disorder, prevention, Indigenous-led, culturally-centered, community-based, strengths-based, social and structural determinants of health

## Abstract

In Australia, fetal alcohol spectrum disorder (FASD) is a largely hidden disability that is currently under-recognized, under-resourced, and under- or misdiagnosed. Unsurprisingly, efforts to prevent FASD in urban Aboriginal and Torres Strait Islander communities are lacking. Further, mainstream approaches are not compatible with diverse and distinct Aboriginal and Torres Strait Islander ways of approaching family, pregnancy, and parenting life. To support the creation of culturally appropriate urban Aboriginal and Torres Strait Islander FASD prevention strategies, we sought to understand local perspectives, experiences, and priorities for supporting healthy and alcohol-free pregnancies. Using a narrative methodology, we undertook research yarns with eight female and two male community participants. Data were analyzed using a narrative, thematic analysis and guided by an Indigenist research practice of reflexive listening. Participant yarns provided important insights into local urban Aboriginal and Torres Strait Islander cultural, social, and structural determinants that support family and child health, alcohol-free pregnancies, and the prevention of FASD. The results provide critical guidance for Indigenizing and decolonizing FASD prevention strategies to support culturally safe, relevant, and strengths-based services. This approach has critical implications for all health and social professionals and can contribute to Aboriginal and Torres Strait Islander peoples’ justice, recovery, and healing from colonization.

## 1. Introduction

Aboriginal and Torres Strait Islander communities, regardless of geographical location, practice family life outside of normative Euro-Australian approaches [1]. While recognizing the intrinsic cultural diversity among Aboriginal and Torres Strait Islanders (hereafter respectfully referred to as Indigenous) communities across Australia, important continuities exist across Indigenous approaches to child raising and family life that are present in urban, regional, and remote communities. While Euro-Australian families tend to be centered around the nuclear family, Indigenous family life is commonly integrated into broader family and community connections—with extended biological and non-biological family members playing important roles in parental support and child raising [1,2]. 

Indigenous cultural practices continue in various capacities despite experiencing significant disruptions to intergenerational transmission from historical and contemporary colonization, for example, through the dispossession of culture and land, widespread massacres, the forced removal of children during the Stolen Generations period, and enduring impacts of institutionalized racism, which entrench socioeconomic disadvantage and inform State child removal policies [3,4,5]. In a demonstration of resilience, Indigenous cultural approaches to family remain significant aspects of community life [2,6]. Cultural family practices support parental and child health and wellbeing, including the prevention of fetal alcohol spectrum disorder (FASD) in urban communities [2,7]. 

FASD is a complex neurodevelopmental disorder that arises from prenatal alcohol exposure and is a leading cause of non-genetic disability in Western countries [3]. FASD prevalence research is currently lacking in Australia; however, comparable countries such as Canada estimate FASD to be at least 2.5 times more prevalent than autism spectrum disorder (1.52% prevalence) [8]. In Australia, FASD is a largely hidden disability, commonly under-recognized, under-resourced, and under- or misdiagnosed [3,9,10,11]. Not surprisingly, efforts to prevent FASD in urban Indigenous communities are lacking, and the complexities of prenatal alcohol use are poorly understood [3,10,12,13]. 

Centering urban Indigenous healthcare priorities is pivotal in the development of urban Indigenous FASD prevention services. In South-East Queensland, where this study took place, an overarching healthcare priority is the delivery of strengths-based, culturally safe services [14,15,16]. Such approaches challenge pervasive deficit discourses of Indigenous community health that have narrowly focused upon family dysfunction rather than function—whereby Indigenous cultural distinctiveness is portrayed as a liability rather than a source of strength and wellbeing [17]. Such deficit narratives persist in the absence of contextual understandings of the broader social and structural milieu that Indigenous families operate within and maintain systematized racial discrimination and inequity that perpetuates social and health disparities [4]. Instead, Indigenous cultural and strengths-based approaches to health focus on those factors that support family health, resiliency, wellbeing, and healthcare safety [2]. 

To support the creation of culturally appropriate and relevant urban Indigenous FASD prevention strategies, we sought to understand local Indigenous perspectives, experiences, and priorities for supporting healthy and alcohol-free pregnancies.

## 2. Materials and Methods

### 2.1. Study Setting and Design

This qualitative study took place in Yuggara Country, with the Indigenous community of Inala, a south-western suburb of Brisbane. The Inala Community Jury for Aboriginal and Torres Strait Islander Health Research [18] identified addressing FASD as a research priority and provided support for research to proceed to inform future services at the Southern Queensland Centre of Excellence in Aboriginal and Torres Strait Islander Primary Health Care (COE) (the Inala Community Jury is the COE’s community research advisory group, consisting of local Indigenous community members who oversee all research conducted through the service). The COE is a Queensland Government funded Indigenous primary health care service that operates in Inala [19]. 

To support a culturally appropriate approach to our investigation, we used a narrative research methodology [20,21]. This conversational and storytelling approach is valued as being compatible with Indigenous knowledge practices and oral traditions [22], and supported our use of the Indigenous research method of yarning [23,24]. Indigenous yarning is a cultural and relational form of conversation—supporting connectedness, situatedness, and transparency through shared dialogue and deep listening [23,24,25]. This approach enables participants to share in-depth insights from their lived experiences and cultural knowledge on their own terms [24]. Importantly, Indigenous yarning is a decolonizing approach that addresses inherent researcher–participant power imbalances and connected concerns of trust and cultural safety [26]. To foster a rich understanding, we drew upon the Indigenist method of Dadirii—a practice of deep listening for what is being shared and not shared [27]. To help prevent harm, we practiced critical reflexivity in our interpretation of participants’ stories and research conduct. We approached this by being attentive to our personal cultural, political, and social biases and positions of power as researchers. We fostered personal accountability and transparency in our research intensions and conduct through team reflections on our interpretations of participants’ stories. We explore this process further in Section 2.2 [28,29]. 

### 2.2. Indigenous Community Governance, Ethics, and Researcher Reflexivity

Prior to commencement, we sought feedback and consent for the project from the Inala Community Jury [18]. Ethical clearance was also obtained from Metro South Human Research Ethics Committee (HREC/19/QMS/54964), and all participants provided written informed consent prior to research yarning. 

The project’s viability was due to the established community relationships and trust that the research team had fostered over years of working in Inala. Aboriginal Research Officer S.E. has worked within her community over the past decade and had established relationships with all study participants. S.E. guided all aspects of the project to ensure its safety and appropriateness for the community. Researchers D.A. and V.L. are both non-Indigenous and therefore do not have a lived understanding of Indigenous family life, culture, and experiences of systematic historical and contemporary racism and discrimination that is part of Indigenous life in Australia. Therefore, they relied upon S.E.’s cultural guidance and oversight in all aspects of the research. D.A. is the former COE Research Director (2010 to 2020) and was already known to most of the participants and Jury members. V.L. is a former COE Research Officer (2016–2018), where she worked closely with COE Indigenous staff, as well as the Jury and community members on several projects. 

### 2.3. Participants and Data Collection

We used a critical sampling strategy [30] to enable us to include a diversity of community participants representing different age groups and genders, who were known by S.E. and D.A. to have rich insights into the research topic, given their personal family and professional community experiences. S.E. approached ten members of the Inala Indigenous community who confirmed they were comfortable to yarn about alcohol use, pregnancy, and FASD prevention, and consented to participate. Research yarns were conducted by two authors (D.A. and S.E.), who used an unstructured approach, enabling participants’ control over the telling of their stories [22,25]. As a starting point, D.A. and S.E. debriefed the participants about the topic they wished to yarn about, concerning alcohol use, pregnancy, and FASD. The researchers then followed the lead of participants, occasionally asking research-related questions if the yarns needed prompting. The yarning process weaved in and out of social and research yarning [31], allowing natural space for affirming and building connectedness between the researchers and participants, as well as understanding each other’s relationship to the local community. 

Research yarns occurred between late 2019 and early 2020. Recruitment ceased with the onset of COVID-19 lockdowns and redeployment of research staff and did not recommence as analysis revealed sufficient depth and diversity of themes [32]. Yarns ranged from 30 to 75 min, and participants received an AUD 30 gift voucher in recognition of their expertise and research contribution. With participants’ consent, yarns were audio recorded and transcribed verbatim by an external organization. To protect participants’ privacy, S.E. deidentified transcripts before analyzing and sharing findings with the broader research team. 

### 2.4. Data Analysis

Data were analyzed using narrative, thematic analysis [20,33], whereby overarching story themes were identified within each participant’s story, with the aim of keeping participants’ voices in context. Following the Dadirii method and personal practices of critical reflexivity was crucial to this process, as was our consideration of the socio-cultural and historical contexts of participants’ lives [26,29]. 

To identify themes, V.L. immersed herself in the data by listening to the audio recordings and reading and re-reading the transcripts. Using NVivo 12 [34] for data management, V.L. identified an overarching core narrative that reflected a life course storyline and themes. These were reviewed with S.E. and D.A. and revised based on discussions until agreement was reached that the story themes appropriately reflected participants’ sentiments shared. S.E. drew on her personal, community, and cultural knowledge to ensure the data analysis had integrity with participants and her community. Upon completion, V.L., D.A. and S.E., presented the findings back to the Inala Community Jury (22 July 2022), who approved the finalization of the research and its publication without request for further changes. 

## 3. Results

Eight female and two male members of the Inala Indigenous community participated in this research. Participants ranged in age from their mid-20s to 70s. They held multiple social and familial roles in the community as mothers, grandmothers, aunties, fathers, and grandfathers, with the majority having worked in social and health service and volunteer roles. Participants’ narratives tell of distinct urban Indigenous ways of knowing and being across key life stages that lift-up cultural approaches to pregnancy and parenting, and support maternal and child health and wellbeing. Woven throughout yarns were participants’ knowledge, experiences, and cautionary views around navigating alcohol use in social and family life, including during pregnancy and parenting. Four main story themes were identified: community life, parenting and family life, pregnancy time, and reflections on the role of health services in supporting healthy and alcohol-free pregnancies. 

### 3.1. Community Life

Participants described their community as proud, strong, and connected, representing a diversity of Indigenous Nations. Particularly, the visibility of the Indigenous community was described as affirming place, belonging, and cultural safety for those who live and visit the area. 

“So I think for a lot of black people and a lot of people of other cultures, they feel safe because they are in a place where they can see themselves and they can see you know that—yeah, they can see themselves and see their culture represented throughout the community…”(p. 2)

Contributing to this visibility, several local Indigenous organizations were identified as valued community pillars, connecting families and increasing the community’s access to culturally relevant services. Yet, at the heart of participants’ stories were the valued community relationships that shaped their family lives. Here, Indigenous yarning was highlighted as an important facilitator of community connectedness, caring, and support. 

“Like, I just came to here [to Inala] … I didn’t have [support]—yeah, somebody would tell you something… it’s just all that community yarning. Yeah, it’s so powerful… because that’s one thing I really get out of Inala… there is always someone there that can maybe give you some information that you can go and do… but Inala itself has that, because people do… they want to help you…”(p. 4)

Participants affirmed an Indigenous approach to family that includes biological and non-biological members. The proximity of family members and regularity of family gatherings were valued, as was a sense of shared responsibility for caring for one another. 

“And everyone is like one big family and we all look after each other’s kids through good and bad times… and we know that we have that here.”(p. 1)

When participants considered the role of alcohol in family and community life, togetherness was the foremost driver of gatherings. Similar to broader Australian society, for many participants, alcohol was a normal part of social life. However, some reflected on times past when heavy alcohol use was prevalent in the community, along with life chapters characterized by trauma, stressful relationships, and circumstances. 

“And there would be those that just drink to get drunk; they’re either stressed out, or just need to let it go, leave the week behind or whatever… I think it’s just a temporary Band-Aid. It’s just that quick little buzz. And I think they’re trying to cope… They’re just getting that quick little—probably that time where they’re not thinking about that stress, and being weighed down with whatever’s going on in their lives.”(p. 8)

When identifying underlying factors influencing heavy alcohol and other substance use, participants highlighted intergenerational stressors due to structural disadvantage and discrimination that the community continues to experience. 

“Inala’s good. I like living here. I’m proud of Inala. It’s just that we’ve had our fair share of problems: drug use, unemployment, police harassment and discrimination. And a lot of that is still happening today. Nothing much really changed… I’ve experienced it [police harassment] personally over the years—and my family and other members in the community that we know…”(p. 3)

### 3.2. Parenting and Family Life

Several participants described the care and assistance received from biological and non-biological family members in the community as foundational to their child raising. Connected to this, learning about parenting was emphasized as commencing in younger years through being included in the care of younger siblings and cousins. 

“… my mum would have maybe four of her siblings living with us and they all had babies and we were all older, so we had to help with those babies and that’s where we were taught all our little things [about child raising] that she was taught—so they all teach us, all our aunties as well. Not just our mother, they were all taught the same but yeah, our aunties as well would teach us.”(p. 7)

Indigenous ways of caring for one another, and particularly children in the community, were an important feature of the community service roles that many participants held. Participants shared how their community relatedness and connections helped them be aware of the unique needs and challenges that young ones may be facing. Understanding family dynamics and if alcohol and other substances were used during pregnancy also helped community workers draw connections with some children’s behavioral challenges and tailor care accordingly. 

“… I was working as the administrator and a court support liaison person. And the people [parents of clients] that I went to school with, that I know, were, and still are, alcohol drinkers. And they drank through pregnancies, they drank through sadness, they drank through happiness, they did. [I knew these clients struggled with understanding bail conditions] It was sort of—it was in the body language, it was in the fact that there were things that I would say that I know myself, I would question… So I knew I had to try and explain a little bit more… Yeah, I don’t know why. I put it down to the fetal alcohol syndrome…”(p. 5)

When yarning about ways of de-centering alcohol use norms in the community, older participants described their own reconciliation with alcohol use across their lives—stories that were spoken with a sense of empowerment. To help teach younger ones about their choices around alcohol, one participant made a point of role modeling alcohol abstinence or low-level consumption at gatherings where drinking was expected. 

“But I make sure that when we do have twenty-first [birthdays] and family functions… I always try to tell my nieces and nephews that are drinking, I’m still having fun, I’m the first one up on the dance floor, I’m the last one up on the dance floor. I can still have fun. And they’re always like, “Aunty, are you drinking?” I’m like, “No, I don’t have to. I’m good. I’m going to be fine tomorrow; you guys are going to be hungover,” and stuff like that.”(p. 8)

Finally, participants shared family approaches to protecting children from alcohol use exposure. For some, alcohol was excluded from children’s gatherings and moderated at family and social gatherings when children were present. Related to this, the role of grandparents in families was highlighted by one participant (p. 3). She described how she and other grandparents in the community worked to protect their grandchildren if parents were using alcohol and other substances to cope with stressors. While not without personal challenges, pride in protecting children, not only from domestic harm, but also from state intervention, was palpable. 

“Due to drugs and alcohol. Whereas the parents aren’t able to focus or function, so in order not to see our grannies [grandchildren], our kids, go into foster home or—we step in and we take them, look after them. None of us care much for Child Safety… Our main concern is our children. Get them out of harm’s way… That just goes way, way back… It’s just something we all do. It must happen in the white community as well sometimes, but not very often. Not as much as within our Murri [Murri is a demonym for Aboriginal Australians of modern-day Queensland and north-western New South Wales, www.wikiwand.com] community, I would say.”(p. 3)

### 3.3. Pregnancy Time 

When yarning about pregnancy, female participants shared about the primacy of their Indigenous ways of intergenerational learning about pregnancy, self-care, and parenting.

“We take our knowledge from our mothers, so, we’re going to believe our mothers before we believe you [non-Indigenous healthcare professionals]. For any of those nurses or anything—we feed our kids like this and we bath our kids like this and that’s from our mothers. Our mothers have handed that to us. Her mother has handed that to her. So, that’s the cycle again…we believe what our mother says, how to look after ourselves. We’ve always believed in what we were supposed to do when we were pregnant.”(p. 7)

Some participants stressed barriers to learning about and supporting women’s and child health and alcohol-free pregnancies when misinformation is passed through Indigenous learning channels. 

“It comes from what’s been passed down to you as well, so there are people that may not get that message that drinking during pregnancy is not good for the baby. So they might not then be able to make a decision that’s best for the baby because they don’t have anybody guiding them with that and particularly if you don’t get that from your family, and if your partner doesn’t understand that as well, then you are not going to be able to get that support from the partner.”(p. 2)

Different ideas about the risks of alcohol use during pregnancy are unsurprising when considering the changing pregnancy health guidelines across different generations—a point several participants touched upon. 

“Cause when I was growing up, it was normal to see pregnant women drink and smoke. Because it wasn’t out there like the effects that alcohol, drugs, and smoking does to a child. It was like having—like, with seatbelts, you didn’t have seatbelts back in them days… they didn’t have that—the information out there and the research… it just wasn’t there.”(p. 7)

One participant (p. 2) also pointed to the impact of the Stolen Generations in disrupting the intergenerational transmission of pregnancy and parenting knowledge and traditions. When reflecting on current alcohol use norms in the community today, several participants considered that only a minority of women might use alcohol after pregnancy recognition, usually in response to life stressors, to help “calm themselves down” (p. 10). Furthermore, there were conflicting views among older participants that there exists greater awareness these days about prenatal alcohol use risks. However, there was shared agreement that a lack of awareness and understandings about FASD was common. 

“And I never drank through my pregnancies because it’s not right and then to this day I don’t see a lot of people pregnant drinking [in my social circles] but I do see a lot of younger ones pregnant drinking on a regular basis, getting drunk, being pregnant. And they don’t like people telling them that they can’t drink. Or Facebook status is about them, people whinging to them about you shouldn’t drink while you’re pregnant. Why, what’s it going to do? They don’t know the effects of it.”(p. 1)

Several participants made special mention of partners and their role in supporting women’s alcohol reduction or abstinence during pregnancy, although significant barriers were noted for mothers that were solo parenting. 

“So, [the father] would just be that support system. A quiet support system. Yeah. You know, my mother knows best. That’s the way we do it… and I just think that support [from the father]—and that’s even, like, washing, feeding, all that, like, washing the clothes, the babies, newborn clothes before they come home.”(p. 7)

Having a supportive partner that refrained from excessive alcohol use during pregnancy and parenting was stressed by participant (p. 4) as vital for supporting women’s health, capacity, and wellbeing.

“It’s so massive. Like, they have got a massive responsibility to support the women. And go and be there, through that whole journey with them. Because having two children to two different men, the difference in having the supportive man, that goes to your appointments, and goes with you, is massive. But then if you’ve got one that’s out partying, yeah, you’re on your own.”(p. 4)

### 3.4. The Role of Health Services in Supporting Healthy and Alcohol-Free Pregnancies 

All participants reflected upon the role of health services in supporting healthy and alcohol-free pregnancies and FASD awareness. Acknowledging Indigenous approaches to family and pregnancy, participants agreed on the need for health and social programs that target the whole-of-community and not just mothers. 

“Because the more knowledge, and more support that you have around you, that’s not going to harm, it’s going to empower people.”(p. 4)

Supporting an inclusive approach, one participant acknowledged the lack of opportunity that older community members have had to learn about risks associated with alcohol use during pregnancy and FASD. 

“Yeah, that’s right because if we don’t like educate our Elders. They didn’t have that opportunity to be educated and they can set that through their families… So I think just promoting it at all age levels, especially our Elders.”(p. 7)

Several participants stressed the importance of services developing culturally appropriate and safe messages. To facilitate this, the quality of relationships that healthcare practitioners can establish with community members was considered essential for effective message delivery. 

“And, depending on your relationship with your doctor and your health service as well, it’s how that message [about alcohol use risks during pregnancy] is then given to that woman to then be able to make a decision around their behaviors and how they actually care for that pregnancy.”(p. 2)

Participants urged healthcare professionals to avoid using authoritative messages and tones with clients and stressed the negative impact “preaching” to Indigenous people about alcohol use risks. Rather, to support culturally safe learning, the importance of Indigenous health workers delivering messages was emphasized, along with incorporating Indigenous perspectives in messaging resources. 

“It’s the worst thing to do to a black person or anyone, I think, is preach. Making people aware is a bit different. Maybe posters advertising it but coming from an Aboriginal view it might be a bit different than coming from a white man view… from the beginning it’s how do you ask someone these questions without offending someone so that would be interesting with this project.”(p. 1)

Connected to this, one participant cautioned against health practitioners’ use of simple abstinence-only messages with women and their supporters (p. 8). Rather, they stressed that families want to be properly informed so that they understand the reasons behind alcohol-related advice. 

Finally, younger parents were described as being particularly resistant to pregnancy health and alcohol abstinence messages from white healthcare professionals. Rather, one participant noted the increasing role of social media in young Indigenous women’s pregnancy self-care learning. 

“So I think some of our young mums would see some of these ones on social media influencers and that, and probably, maybe, inspiring or striving to be similar…”(p. 8)

## 4. Discussion

Participants’ stories provide important insights into distinct local urban Indigenous cultural, social, and structural determinants that can support family and child health, alcohol-free pregnancies, and prevent FASD. The results highlight the centrality of place, community, culture, and family for supporting women’s health during pregnancy, along with some of the stressors that can be present for Indigenous families. Across participants’ stories, the place and impact of prenatal alcohol use varied considerably, depending on life stage, exposure to changing alcohol use norms over time, and the presence or absence of social and structural supports and stressors. 

While participants’ stories are embedded in distinct geographical, historical, and cultural contexts, their messages are resounded in broader Indigenous affirmations of the holistic factors that support community health and wellbeing, as well as key barriers to these [2,6]. The breadth of holistic Indigenous health determinants exists outside the narrow scope of mainstream public health and medical models of care that inform typical FASD prevention approaches [35]. In response to this distinctiveness, the current results support Gonzales et al.’ [35] proposal that a restructuring of mainstream FASD prevention approaches is required to foster culturally appropriate and relevant services for Indigenous families and their communities. 

### 4.1. Centring Indigenous Knowledge, Culture, and Contexts in FASD Prevention

In the restructuring of FASD prevention efforts for urban Indigenous communities, the importance of culturally centered, Indigenous-led, community-based, and contextualized approaches cannot be understated [2,35,36,37,38,39]. These approaches enable service providers to be grounded in Indigenous conceptions of health and wellbeing that are holistic, relational, and highly relevant to Indigenous lives and ways of ‘doing’ family [6,35]. Indigenous-centered approaches foster cultural safety and service accessibility for community members through centering positive and strengths-based cultural approaches that have supported Indigenous families’ and community wellness and resilience across generations [2,6]. As such, Indigenous-centered approaches are a critical antidote to the continued perpetuation of the colonial discourse of Indigenous deficiency [17]. This discourse has fueled negative perceptions of Indigenous mothers as irresponsible and incapable since the continent’s invasion—a perception that gave rise to the forced removal of Indigenous children through the Stolen Generations [5]. The effects of the Stolen Generations are ongoing; as one participant in our study highlighted, intergenerational Indigenous knowledge around pregnancy, parenting, and self-care was disrupted in many families. Today, the deficit colonial perception continues through the over-surveillance of Indigenous mothers and the disproportionately high involvement of child protection agencies in Indigenous family life [5]. 

Working outside of the deficit paradigm, an Indigenous health determinants approach to FASD prevention addresses the contexts and complexities of Indigenous women’s lives in compassionate, positive, and non-stigmatizing ways [37,40]. Indigenous-led FASD prevention programs in Australia, USA, and Canada, while still emerging, have demonstrated this approach through the use of harm-reduction and needs-based strategies that work with the lived realities of women’s lives [40,41,42]. Alongside support for alcohol and other substance use, strategies have included support for domestic violence, transport, housing, and food security, as well as social and cultural connectedness [40,43]. Such holistic approaches help ameliorate barriers to alcohol and other substance use reduction and abstinence and work from the premise that supported women have healthier pregnancy outcomes and greater capacity for parenting. This was particularly apparent in our study, where some participants described using alcohol and other substances to cope during life chapters shaped by stress and unsupportive relationships. Importantly, holistic approaches can work within Indigenous women’s family structures by not only recognizing the role of partners in influencing women’s health behaviors but also the role of extended family (biological and non-biological) and Elders and other community members [40]. These harm-reducing and family-orientated approaches depart from mainstream FASD prevention models that are typically individually focused, stressing women’s total abstinence from alcohol during pregnancy and constructing FASD as 100% preventable without contextual understanding [37,43,44].

An Indigenous health determinants lens also recognizes the relationship between intergenerational trauma, violence, and Indigenous women’s health behaviors [40,45]. This orientation is increasingly recognized as foundational to FASD prevention efforts for Indigenous communities [35,39,46,47,48]. As study participants highlighted, women’s experiences of intergenerational and present-day trauma and domestic violence had direct impacts on their use of alcohol and other substances during pregnancy and parenting years. 

In Canada, the causal relationship of colonization, specifically the impact of the residential school system, with the prevalence of FASD in Indigenous communities has been recognized as being of critical importance and was included in the nation’s Truth and Reconciliation Commission action plan [39]. This landmark recognition heralds a formal departure from negative stereotypes surrounding Indigenous communities, mothers, and alcohol use, which have damagingly problematized Indigenous cultures and communities [37]. In response, the Commission instead problematized historical and contemporary mechanisms of colonization and called for decolonizing and Indigenizing pathways of action that support Indigenous peoples’ recovery, healing, cultural strength, and resiliency [39,45]. From this work, we see how Indigenizing and decolonizing FASD prevention is a social justice imperative that implicates all professionals working in this space.

### 4.2. Progressing Indigenous Approaches to FASD Prevention 

In Australia, FASD and its prevention, although gaining recognition, continues to be poorly understood and underserviced in mainstream health services [9,49]. Participants in our study discussed the lack of culturally safe and trusted health messaging for Indigenous people in mainstream health services, as well as the increased occurrence of young people accessing health information through social media, irrespective of its veracity. In supporting the restructuring of mainstream medical models for preventing FASD and tailoring efforts to urban Indigenous contexts, health and social service practitioners play a pivotal role. As a starting point, stigma has been identified as a major barrier to practitioners implementing FASD prevention [43,44,50]. For example, practitioners are hesitant to routinely ask all pregnant women/people about alcohol use for fear of jeopardizing patient relationships, as well as the occurrence of selective inquiry informed by practitioners’ personal judgments about who may or may not be consuming alcohol [43,51]. Additionally, the predominance of individual behavioralist approaches for addressing FASD reflects a lack of consultation and collaboration with Indigenous communities themselves [52]. Not surprisingly, a failure to address relevant contextual factors through meaningfully involving partners, parents, and extended family and community members in FASD prevention approaches has been identified [52]. 

In response, we encourage the centering of Indigenous leadership and knowledge, and collaboration with communities to ground FASD prevention efforts in local culture. By doing so, health and community services and non-Indigenous practitioners can be better informed to support local priorities and cultures. Practitioners will also be better positioned to foster culturally safe, relevant, and strengths-based FASD prevention approaches. Participants in this study urged practitioners to avoid preaching and authoritative messaging and tones in their prevention work, approaches that are symptomatic of a deficit lens towards Indigenous people. Rather, we urge services and practitioners to commit to self-reflexive practices to challenge conscious or unconscious biases and narratives towards Indigenous people. Also, to be cognizant of the limitations and potential negative impact of Western concepts of health and wellbeing that invariably shape practitioners’ provision of care [6,53,54]. 

### 4.3. Strengths and Limitations 

The study contributes to an emerging body of qualitative literature that prioritizes learning from local Indigenous communities to inform the cultural and contextual tailoring of FASD prevention services. We see this approach as vital. First, centering community voices supports the creation of strategies that are highly relevant to local Indigenous family life, culture, needs, preferences, and priorities. Second, a localized study focus enabled us to avoid homogenizing the many distinct urban Indigenous communities across Australia. As such, the nature of the findings precludes the generalization of participants’ stories to other urban communities. The findings do, however, share important insights for healthcare services and professionals who seek to progress urban Indigenous FASD prevention strategies that are culturally centered, Indigenous-led, and community-based. We attribute the depth of insights gained in this study to the quality of pre-existing researcher-participant relationships that we believe enabled the yarning process and better positioned the researchers to understand the nuances of participant’s stories. While the findings provide an important orientation to tailoring FASD prevention to local community cultures and priorities, this research does not replace the need for ongoing community collaboration and developmental evaluation to ensure the appropriateness and cultural safety of future FASD prevention services. 

## 5. Conclusions

To support the creation of culturally appropriate and relevant FASD prevention services for urban Indigenous people, we sought to understand local Indigenous perspectives, experiences, and priorities for supporting healthy and alcohol-free pregnancies. Participant yarns highlighted the importance of place, community, culture, and family, and some of the structural factors influencing women’s health behaviors during preconception, pregnancy, and parenting years. Insights from the yarns align with an Indigenous health determinants perspective and emerging Indigenous approaches to FASD prevention. Together, they call for health interventions that are Indigenous-led, culturally centered, holistic, relational, and strengths-based. The findings offer a critical road map for reorientating mainstream FASD prevention work beyond the narrow confines of individualized behavioral change approaches that are prevalent in public health and medical models. Rather, Indigenous approaches to FASD prevention support harm-reduction strategies that are tailored to the lived realities and priorities of urban Indigenous women and their families. This reorientation has important implications for all professionals who work in this space and contributes to the broader movement toward Indigenous justice, recovery, and healing from colonization. 

## Data Availability

The data that supports the findings of this study are not publicly available, but are available from the University of Queensland via correspondence with the author, provided appropriate ethical and community approvals are obtained.

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
