# Peer review of "“Our Mothers Have Handed That to Us. Her Mother Has Handed That to Her”: Urban Aboriginal and Torres Strait Islander Yarning about Community Wellbeing, Healthy Pregnancies, and the Prevention of Fetal Alcohol Spectrum Disorder"

_ijerph, 2023, doi:10.3390/ijerph20095614_

Round 1
Reviewer 1 Report
This is an important study that is beautifully written. The importance of cultural knowledge and in programs that promote care and prevention cannot be overstated, and the authors do a wonderful job of marking this point trough compelling narratives or yarns. My only suggestion is to expand on the demographics. Because this is a study that explores how change is occurring it would be valuable to know how old the individuals make these comments are. Noting any differences across age groups in the body of the text would strengthen the analysis and conclusions. I am making the same comments to the editor.
Author Response
This is an important study that is beautifully written. The importance of cultural knowledge and in programs that promote care and prevention cannot be overstated, and the authors do a wonderful job of marking this point trough compelling narratives or yarns. My only suggestion is to expand on the demographics. Because this is a study that explores how change is occurring it would be valuable to know how old the individuals make these comments are. Noting any differences across age groups in the body of the text would strengthen the analysis and conclusions. I am making the same comments to the editor.
We are grateful for your positive feedback and suggestions for our paper.
Due to the study’s small sample size and the closeness of the local community, we chose to refrain from directly identifying participants by their demographic characteristics to maintain their privacy and confidentiality. However, to better describe the participants we have made the following adjustment to lines 169 to 173:
“Eight female and two male members of the Inala Indigenous community participated in this research. Participants ranged in age from their mid-20s to 70s. They held multiple social and familial roles in the community as mothers, grandmothers, Aunties, fathers, and grandfathers, with the majority having worked in social and health service and volunteer roles.”
Reviewer 2 Report
Review of: “Our mothers have handed that to us. Her mother has handed 2 that to her”: Urban Aboriginal and Torres Strait Islander yarn-3 ing about community wellbeing, healthy pregnancies, and the 4 prevention of fetal alcohol spectrum disorder.
Thank you for asking me to review this manuscript. I really enjoyed reading it and learned a lot. The writing is very clear and easy to follow. This paper is a great addition to the existing literature. I have some minor points to share, which are below.
Regarding these two sentences: Abstract: In Australia, fetal alcohol spectrum disorder (FASD) is a largely ‘hidden’ disability, that is currently under-recognized, under-resourced and under- or misdiagnosed. (lines 15-16; 60)
I’m not sure why hidden has quote marks around it. It doesn’t seem like they are needed.
Regarding this sentence: This approach has critical implications for non-Indigenous health professionals and can contribute to Aboriginal and Torres Strait Islander peoples’ justice, recovery, and healing from colonization. (lines 29-31 and also 536-537)
It seems like these findings have implications for a broader audience than non-Indigenous health professionals. I suggest broadening this as it seems like you learned perspectives and information that would be helpful to a broader audience. It looks like your discussion provides implications for a broader audience.
It would be helpful to include a citation for this sentence: Cultural family practices support parental and child health and wellbeing, including the prevention of fetal alcohol spectrum disorder (FASD) in urban communities. (lines 53-54)
The paragraph on lines 63-75 is lovely. It is clear to see the care you took with your writing.
Regarding this sentence: To help prevent harm, we practiced critically reflexivity in our interpretation of participants’ stories, and transparency in our research intentions and conduct [25, 26]. (lines 101-103).
I would love to see an additional paragraph that describes what this meant in action. In other words, what did you do to practice critical reflexivity and transparency? If you feel that this is explained below, you could add something like, “which is explained in section xx below.”
Regarding these sentences: SE guided all aspects of the project to ensure its safety and appropriateness for the community. Researchers DA and VL are both non-Indigenous and therefore do not have lived understanding of Indigenous family life, culture, and experiences of systematic historical and contemporary racism and discrimination that is part of Indigenous life in Australia. Therefore, they relied upon SE’s cultural guidance and oversight in all aspects of the research. (lines 113-118)
I really appreciate this.
Regarding this sentence: Using a critical sampling strategy [27], SE approached ten members of the Inala Indigenous community who were comfortable in yarning about alcohol use, pregnancy and 125 FASD prevention. (lines 124-125)
I’d love to see a sentence added that defines critical sampling strategy and a description of how you all knew they were comfortable beforehand. Is this a close-knit community where anyone would know this? Did the participants participate in another program that enabled you to know this?
It would be nice to have a little more information about the participants as well. Were they parents, grandparents, approximate age range, etc.
I’d like to know a little more about the yarning process and be able to picture what happened. Were there specific questions asked of each participant or an overall question asked in the beginning? For example, were participants asked to talk about FASD specifically or pregnancy more generally or both? A few sentences to place the reader there to better understand would be helpful. Yarning is likely a new term to some readers, so to share for example, what does it mean to yarn about alcohol use, pregnancy and FASD prevention, would be good.
Regarding this phrase: analysis revealed sufficient depth and diversity of themes (line 133)
Is this something that is recognized in the literature as when more interviews are not needed? A citation would be helpful.
Did the community jury (line 153) have any changes to what was presented or was there any discussion?
Regarding this sentence: Alcohol featured moderately in most stories. (lines 197-198)
I’m not sure what being featured moderately means. Please describe a bit more.
Regarding: [I knew these clients struggled with understanding bail conditions] (line 240)
I’m not sure why this is in brackets.
Regarding: We take our knowledge from our mothers, so, we’re going to believe our mothers before we believe you. For any of those nurses or anything (lines 281-282).
I got a little confused here. Is “you” the nurses who are non-Indigenous? I’m not sure what “for any of those nurses” means. Maybe add a little more information to clarify.
Regarding section 4.1, the weaving of what you learned into the broader context of published information is really well done. Section 4.3 is clearly written and laid out and provides specific strengths-based helpful information. Great job!
The sections go from 4.1 to 4.3
Section 4.4 is labeled strengths and limitations. Is the last sentence the limitation? It wasn’t clear to me.
I’d like to comment on the title. I love the quote because it is strengths-based and culturally consonant and from one of your participants. However, in the section the quote comes from you discuss that there is also misinformation that is passed along due to colonization, so it’s more complication/contextual (like everything). I leave it to you to decide what to do.
Again, I really enjoyed reading this paper, which will make a nice contribution to the literature.
Author Response
Thank you for asking me to review this manuscript. I really enjoyed reading it and learned a lot. The writing is very clear and easy to follow. This paper is a great addition to the existing literature. I have some minor points to share, which are below.
Thank you for your thoughtful and supportive review and recommendations.
Regarding these two sentences: Abstract: In Australia, fetal alcohol spectrum disorder (FASD) is a largely ‘hidden’ disability, that is currently under-recognized, under-resourced and under- or misdiagnosed. (lines 15-16; 60)
I’m not sure why hidden has quote marks around it. It doesn’t seem like they are needed.
We have taken out these quotation marks as recommended.
Regarding this sentence: This approach has critical implications for non-Indigenous health professionals and can contribute to Aboriginal and Torres Strait Islander peoples’ justice, recovery, and healing from colonization. (lines 29-31 and also 536-537)
It seems like these findings have implications for a broader audience than non-Indigenous health professionals. I suggest broadening this as it seems like you learned perspectives and information that would be helpful to a broader audience. It looks like your discussion provides implications for a broader audience.
We have made the following adjustments to incorporate a boarder audience focus:
Lines 29-31: “This approach has critical implications for all health and social professionals and can contribute to Aboriginal and Torres Strait Islander peoples’ justice, recovery, and healing from colonization.”
Lines 552-553: “This reorientation has important implications for all professionals who work in this space, and contributes to the broader movement towards Indigenous justice, recovery, and healing from colonization.”
It would be helpful to include a citation for this sentence: Cultural family practices support parental and child health and wellbeing, including the prevention of fetal alcohol spectrum disorder (FASD) in urban communities. (lines 53-54)
We have included the following two citations to support this sentence:
Hewlett, N., Hayes, L., Williams, R., Hamilton, S., Holland, L., Gall, A., Doyle, M., Goldsbury, S., Boaden, N. and Reid, N. Development of an Australian FASD Indigenous Framework: Aboriginal Healing-Informed and Strengths-Based Ways of Knowing, Being and Doing. Int. J. Environ. Res. Public Health 2023, 20(6), 5215; 10.3390/ijerph20065215.
Verbunt, E., Luke, J., Paradies, Y., Bamblett, M., Salamone, C., Jones, A. and Kelaher, M. Cultural determinants of health for Aboriginal and Torres Strait Islander people - a narrative overview of reviews. Int. J. Equity Health 2021, 20, pp. 181. 10.1186/s12939-021-01514-2.
The paragraph on lines 63-75 is lovely. It is clear to see the care you took with your writing.
Thank you.
Regarding this sentence: To help prevent harm, we practiced critically reflexivity in our interpretation of participants’ stories, and transparency in our research intentions and conduct [25, 26]. (lines 101-103).
I would love to see an additional paragraph that describes what this meant in action. In other words, what did you do to practice critical reflexivity and transparency? If you feel that this is explained below, you could add something like, “which is explained in section xx below.”
We have made the following adjustments to lines 101-107:
To help prevent harm, we practiced critical reflexivity in our interpretation of participants’ stories and research conduct. We approached this through being attentive to our personal cultural, political, and social bias and positions of power as researchers. We fostered personal accountability and transparency in our research intensions and conduct through team reflections on our interpretations of participants’ stories. We explore this process further in section 2.2 [28, 29].
Regarding these sentences: SE guided all aspects of the project to ensure its safety and appropriateness for the community. Researchers DA and VL are both non-Indigenous and therefore do not have lived understanding of Indigenous family life, culture, and experiences of systematic historical and contemporary racism and discrimination that is part of Indigenous life in Australia. Therefore, they relied upon SE’s cultural guidance and oversight in all aspects of the research. (lines 113-118)
I really appreciate this.
Regarding this sentence: Using a critical sampling strategy [27], SE approached ten members of the Inala Indigenous community who were comfortable in yarning about alcohol use, pregnancy and 125 FASD prevention. (lines 124-125)
I’d love to see a sentence added that defines critical sampling strategy and a description of how you all knew they were comfortable beforehand. Is this a close-knit community where anyone would know this? Did the participants participate in another program that enabled you to know this?
To clarify our approach, we have made the following adjustments on lines 129 to 134:
“We used a critical sampling strategy [30] to enable us to include a diversity of community participants representing different age groups and genders, who were known by SE and DA to have rich insights into the research topic, given their personal family and professional community experiences. SE approached ten members of the Inala Indigenous community who confirmed they were comfortable to yarn about alcohol use, pregnancy and FASD prevention and consented to participating.”
It would be nice to have a little more information about the participants as well. Were they parents, grandparents, approximate age range, etc.
Due to the study’s small sample size and the closeness of the local community, we chose to refrain from directly identifying participants by their demographic characteristics to maintain their privacy and confidentiality. However, to better describe the participants we have made the following adjustment to lines 169 to 173:
“Eight female and two male members of the Inala Indigenous community participated in this research. Participants ranged in age from their mid-20s to 70s. They held multiple social and familial roles in the community as mothers, grandmothers, Aunties, fathers, and grandfathers, with the majority having worked in social and health service and volunteer roles.”
I’d like to know a little more about the yarning process and be able to picture what happened. Were there specific questions asked of each participant or an overall question asked in the beginning? For example, were participants asked to talk about FASD specifically or pregnancy more generally or both? A few sentences to place the reader there to better understand would be helpful. Yarning is likely a new term to some readers, so to share for example, what does it mean to yarn about alcohol use, pregnancy and FASD prevention, would be good.
To help clarify our approach to research yarning, we have included the following sentences on lines 134 to 142:
“Research yarns were conducted by two authors (DA and SE), who used an unstructured approach, enabling participants’ control over the telling of their stories [22, 25]. As a starting point, DA and SE debriefed the participants about the topic they wished to yarn about, concerning alcohol use, pregnancy, and FASD. The researchers then followed the lead of participants, occasionally asking research related questions if the yarns needed prompting. The yarning process weaved in and out of social and research yarning [31], allowing natural space for affirming and building connectedness between the researchers and participants, as well as understanding each other’s relationship to the local community.”
Regarding this phrase: analysis revealed sufficient depth and diversity of themes (line 133)
Is this something that is recognized in the literature as when more interviews are not needed? A citation would be helpful.
Data sufficiency was met when, upon analysis, we identified in-depth and nuanced of insights regarding our research focus on alcohol use, pregnancy, and the prevention of FASD. Important to this was reaching a representation of a diversity of community perspectives from different age, gender, and personal and professional standpoints. This approach to inductive thematic analysis in qualitative research is well established and described in:
Braun V, Clarke V. To saturate or not to saturate? Questioning data saturation as a useful concept for thematic analysis and sample-size rationales. Qualitative Research in Sport, Exercise and Health 2021;13:201-16. 10.1080/2159676X.2019.1704846.
To strengthen our approach, we have included a citation for this article on line 145.
Did the community jury (line 153) have any changes to what was presented or was there any discussion?
The Community Jury did not require any changes to what was presented. Discussion following the presentation focused upon the importance of the research for informing future efforts to support FASD prevention in the community.
We have made the following changes to clarify the Jury process:
Line 164 to 167: “Upon completion, VL, DA and SE, presented the findings back to Inala Community Jury (22nd July 2022), who approved the finalization of the research and its publication without request for further changes.”
Regarding this sentence: Alcohol featured moderately in most stories. (lines 197-198)
I’m not sure what being featured moderately means. Please describe a bit more.
To avoid confusion with our use of the term ‘moderately’, we have deleted this sentence and feel that doing so does not detract from the findings.
Regarding: [I knew these clients struggled with understanding bail conditions] (line 240)
I’m not sure why this is in brackets.
We provided this additional text in brackets to clarify the context of the quote, whereby the participant was describing their work with supporting young Indigenous people in understanding and adhering to bail conditions.
Regarding: We take our knowledge from our mothers, so, we’re going to believe our mothers before we believe you. For any of those nurses or anything (lines 281-282).
I got a little confused here. Is “you” the nurses who are non-Indigenous? I’m not sure what “for any of those nurses” means. Maybe add a little more information to clarify.
We have made the following adjustments to clarify what is meant by this quote on lines 296 to 297:
“We take our knowledge from our mothers, so, we’re going to believe our mothers before we believe you [non-Indigenous health-care professionals].”
Regarding section 4.1, the weaving of what you learned into the broader context of published information is really well done. Section 4.3 is clearly written and laid out and provides specific strengths-based helpful information. Great job!
Thank you.
The sections go from 4.1 to 4.3
Section 4.4 is labeled strengths and limitations. Is the last sentence the limitation? It wasn’t clear to me.
We have made the following changes to clarify our intention for the final point of our study limitations (lines 532 to 536):
“While the findings provide an important orientation to tailoring FASD prevention to local community cultures and priorities, this research does not replace the need for on-going community collaboration and developmental evaluation to ensure the appropriateness and cultural safety of future FASD prevention services.”
I’d like to comment on the title. I love the quote because it is strengths-based and culturally consonant and from one of your participants. However, in the section the quote comes from you discuss that there is also misinformation that is passed along due to colonization, so it’s more complication/contextual (like everything). I leave it to you to decide what to do.
Thank you for highlighting this complexity, however we have decided to remain with this title. With much consideration this it was chosen because it is from a participant and really captures an aspect of Indigenous ways of learning about pregnancy and parenting in a strengths-based way that was foundational to this study. While the broader context reveals this was not without complexity, we feel that the impact of colonization and systemic inequity does not detract from the importance and strength of this approach to Indigenous ways of knowing. It was our aim for this study to articulate this sentiment, and rather, to look for ways to mitigate the impacts of colonization.
Again, I really enjoyed reading this paper, which will make a nice contribution to the literature.
Thank you again for your positive and constructive feedback.
Reviewer 3 Report
The authors said that to support a culturally appropriate approach to their investigation, they used a narrative research methodology. However, a book has been cited as the source of this method. Since it is not possible to access the book, it is not possible to have information about the method used for the reader.
In the introduction, there is not enough background on why the relationship between the urban Aboriginal and Torres Strait Islander communities and FASD is wanted to be investigated. The authors only mentioned that Aboriginal and Torres Strait Islander communities suffer from discrimination and injustice, and they stated that FASD should be investigated, for which there is insufficient information about its prevalence. However, when it comes to the Methods section, the main reason for this research is revealed.
Author Response
The authors said that to support a culturally appropriate approach to their investigation, they used a narrative research methodology. However, a book has been cited as the source of this method. Since it is not possible to access the book, it is not possible to have information about the method used for the reader.
Thank you for your feedback.
We provided a book reference for our use of a narrative research methodology as much of the foundational text regarding this methodology has been published in this format. However, we have provided an additional reference for this section to help clarify and justify our use of this methodology, which we have cited on line 91:
Polkinghorne DE. Narrative configuration in qualitative analysis. International Journal of Qualitative Studies in Education 1995;8:5-23. 10.1080/0951839950080103.
In the introduction, there is not enough background on why the relationship between the urban Aboriginal and Torres Strait Islander communities and FASD is wanted to be investigated. The authors only mentioned that Aboriginal and Torres Strait Islander communities suffer from discrimination and injustice, and they stated that FASD should be investigated, for which there is insufficient information about its prevalence. However, when it comes to the Methods section, the main reason for this research is revealed.
We respectfully acknowledge your concern that our introduction was not clearly justified, however we feel we have sufficiently articulated its need in the introduction section. As we described, there exists a lack of FASD research in the Australian context, which is particularly pronounced for urban Aboriginal and Torres Strait Islander communities. It is for this reason that FASD researchers in Australia look to comparable countries to describe the seriousness of this issue and the urgency for research that can support prevention work. To further support our investigation, we have cited the following reference on line 63, which highlights the persistent under-representation of urban Aboriginal and Torres Strait Islander communities in health research:
Jennings W, Spurling G, Shannon B, Hayman N, Askew D. Rapid review of five years of Aboriginal and Torres Strait Islander health research in Australia - persisting under-representation of urban populations. Aust N Z J Public Health 2021;45:53-8. 10.1111/1753-6405.13072.
In the methods section we further justify our investigation by describing the research priorities of the Inala Community Jury for Aboriginal and Torres Strait Islander Health Research, and their support for this research to occur.